# Explainable Artificial Intelligence in Neuroimaging of Alzheimer’s Disease

**DOI:** 10.3390/diagnostics15050612

**Published:** 2025-03-04

**Authors:** Mahdieh Taiyeb Khosroshahi, Soroush Morsali, Sohrab Gharakhanlou, Alireza Motamedi, Saeid Hassanbaghlou, Hadi Vahedi, Siamak Pedrammehr, Hussain Mohammed Dipu Kabir, Ali Jafarizadeh

**Affiliations:** 1Student Research Committee, Tabriz University of Medical Sciences, Tabriz 5164736931, Iran; mtayebkhosroshahi@gmail.com (M.T.K.); morsalis@tbzmed.ac.ir (S.M.); sohrabgrxnl@gmail.com (S.G.); alirezamot1380@gmail.com (A.M.); hbsaeed98@gmail.com (S.H.); hadivahedimrg@gmail.com (H.V.); 2Research Center of Psychiatry and Behavioral Sciences, Tabriz University of Medical Sciences, Tabriz 5164736931, Iran; 3Tabriz USERN Office, Universal Scientific Education and Research Network (USERN), Tabriz 5164736931, Iran; jafarizadeha@tbzmed.ac.ir; 4Nikookari Eye Center, Tabriz University of Medical Sciences, Tabriz 5164736931, Iran; 5Faculty of Design, Tabriz Islamic Art University, Tabriz 5164736931, Iran; s.pedrammehr@tabriziau.ac.ir; 6Artificial Intelligence and Cyber Futures Institute, Charles Sturt University, Orange, NSW 2800, Australia; 7Rural Health Research Institute, Charles Sturt University, Orange, NSW 2800, Australia

**Keywords:** Alzheimer’s disease, explainable AI, artificial intelligence, machine learning, deep learning, dementia

## Abstract

Alzheimer’s disease (AD) remains a significant global health challenge, affecting millions worldwide and imposing substantial burdens on healthcare systems. Advances in artificial intelligence (AI), particularly in deep learning and machine learning, have revolutionized neuroimaging-based AD diagnosis. However, the complexity and lack of interpretability of these models limit their clinical applicability. Explainable Artificial Intelligence (XAI) addresses this challenge by providing insights into model decision-making, enhancing transparency, and fostering trust in AI-driven diagnostics. This review explores the role of XAI in AD neuroimaging, highlighting key techniques such as SHAP, LIME, Grad-CAM, and Layer-wise Relevance Propagation (LRP). We examine their applications in identifying critical biomarkers, tracking disease progression, and distinguishing AD stages using various imaging modalities, including MRI and PET. Additionally, we discuss current challenges, including dataset limitations, regulatory concerns, and standardization issues, and propose future research directions to improve XAI’s integration into clinical practice. By bridging the gap between AI and clinical interpretability, XAI holds the potential to refine AD diagnostics, personalize treatment strategies, and advance neuroimaging-based research.

## 1. Introduction

Alzheimer’s disease (AD) poses a significant global health challenge, affecting approximately 50 million individuals worldwide—a number that is expected to increase substantially due to population aging [1]. Beyond its impact on patients, AD places a heavy burden on families, caregivers, and healthcare systems, underscoring the need for early and accurate diagnosis. Currently, the diagnostic process often involves clinical assessments, biomarker analysis (e.g., amyloid-beta, tau), and neuroimaging. However, the phenotypic variability and spatiotemporal complexity of AD biomarkers can complicate interpretation [2].

Recently, artificial intelligence (AI) has proven valuable in analyzing large, multidimensional neuroimaging datasets for the early detection of AD [3,4,5,6]. Machine learning (ML) techniques (e.g., image classification, pattern recognition) and deep learning (DL) models (e.g., Convolutional Neural Networks, Graph Convolutional Networks) have demonstrated high accuracy in identifying AD-related changes, often before clinical symptoms manifest [7,8]. Yet while these advanced models excel at pattern recognition, their opaque decision-making process raises concerns regarding transparency and trust in clinical settings.

Explainable Artificial Intelligence (XAI) is recognized as a key approach to improving model interpretability. By offering human-understandable explanations for AI-driven decisions, XAI can strengthen the clinical utility of automated AD diagnosis. When integrated into neuroimaging workflows—such as MRI- or PET-based analyses—XAI techniques help clinicians pinpoint disease-relevant regions of interest and understand how AI models prioritize specific features (e.g., hippocampal atrophy) [9]. This article aims to review the application of XAI in neuroimaging for AD diagnosis, focusing on interpretability methods (e.g., SHAP, LIME, Grad-CAM, LRP) and how they can improve diagnostic accuracy, biomarker identification, and patient-specific treatment strategies.

## 2. Literature Search Method

We conducted a literature search in IEEE and PubMed Xplore to identify peer-reviewed studies published in English that focus on the application of AI in neuroimaging for AD classification and diagnosis. Additional papers were retrieved using the Google Scholar engine. The search was performed in May 2024 with the keywords and search strategy delineated in Appendix A. We prioritized the most recent publications that introduced novel insights into the field while ensuring that repeated studies were filtered based on quality. We included peer-reviewed cutting-edge original research articles that provided clear research objectives, rigorous methodologies, and detailed explanations of AI-based explainability techniques. The studies selected for review focused on the use of XAI methods in analyzing neuroimaging data for AD classification, critical brain region and biomarker identification, and disease progression assessment. Studies that lacked sufficient technical details, non-peer-reviewed sources, review articles, letters, conference abstracts, and duplicate publications were excluded. Our screening process flowchart demonstrating the article selection process is shown in Figure 1. After the screening phase, relevant data [10,11,12,13,14,15,16,17,18,19,20,21,22,23,24,25,26,27,28,29,30,31,32,33] were extracted into Appendix A.

## 3. AD Overview

AD accounts for 50–70% of all dementia cases, with its progression driven by abnormal protein deposits (amyloid plaques, neurofibrillary tangles) that disrupt neuronal function [34,35]. The disease typically follows a progression from cognitively normal (CN) aging through Mild Cognitive Impairment (MCI) to AD, although some individuals may not neatly fit these stages. Genetic susceptibility (e.g., APOE ε4) and various environmental factors also contribute to the disease’s complexity [34]. With rising life expectancy and an aging population, AD prevalence is projected to increase worldwide, particularly in developing regions, imposing significant healthcare costs [34]. Addressing modifiable risk factors in middle-aged and elderly individuals may help delay or prevent disease onset [34,36].

As biomarkers evolve over the disease course, multimodal diagnostic approaches integrating genetic data, fluid biomarkers, and imaging findings hold promise. In particular, integrating neuroimaging with other data sources (e.g., exosomal microRNAs [37], blood-based biomarkers [38]) can provide a more holistic view of pathophysiological processes.

## 4. Neuroimaging in AD

Neuroimaging is central to diagnosing and monitoring AD, with multiple modalities offering complementary insights. For instance, structural MRI (sMRI) is a non-invasive neuroimaging technique that can detect early changes in brain structures associated with AD [39]. The entorhinal cortex and hippocampus, both part of the medial temporal lobe, are among the earliest brain regions affected by AD [40]. Volumetric measurements of the hippocampus have been used to identify the early stages of the disease [41,42]. Hippocampal atrophy, as evaluated by high-resolution T1-weighted MRI, is the most well-established and validated MRI biomarker for AD. The visual examination of coronal T1-weighted MRI images offers a direct approach to assessing atrophy in the medial temporal lobe. Assessment using visual rating scales demonstrates high sensitivity and specificity in differentiating AD patients from individuals without cognitive impairment and in predicting cognitive decline in MCI. MRI-derived measures of atrophy are considered reliable indicators of disease status and progression for several reasons. Atrophy appears to be an unavoidable and progressively worsening feature associated with neurodegeneration. The spatial distribution of brain tissue loss is closely associated with cognitive impairments, both in cross-sectional and longitudinal studies [43]. sMRI can also assess cortical thickness as a proxy for neuronal loss in the cortex. While sMRI has around 80% accuracy in predicting conversion to AD in MCI cases [44], it lacks molecular specificity and is unable to directly identify AD’s defining pathological features, such as Aβ proteins and neurofibrillary tangles (NFTs) [45]. Functional MRI (fMRI) tracks fluctuations in blood oxygenation and circulation, offering insights into neural activity during cognitive processing or resting states [39]. Compared to healthy controls, individuals with AD show reduced activation in the hippocampus and other medial temporal structures during memory tasks, suggesting compensatory mechanisms in the brain [40]. Resting-state fMRI is a promising biomarker for AD, as it can detect functional brain changes that precede structural alterations. Research indicates that AD affects functional connectivity within the default mode network, leading to reduced connectivity in posterior brain regions and increased connectivity in frontal regions, potentially as a compensatory mechanism [46]. While fMRI provides unique insights into the pathophysiology of AD [47], its application in clinical settings is limited due to variability in the BOLD response and challenges in longitudinal studies [48].

PET has emerged as a pivotal tool in the investigation, diagnosis, and management of AD. In individuals afflicted with this condition, PET serves as a valuable tool for examining alterations in cerebral glucose metabolism, various neurotransmitter systems, neuroinflammation, and pathognomonic protein aggregates, particularly amyloid deposits [49]. Fluorodeoxyglucose (FDG)-PET has been widely employed to diagnose AD, as it measures glucose metabolism in the brain. Studies have demonstrated that FDG-PET can accurately distinguish between AD patients and healthy controls and differentiate between various stages of the disease [50,51]. The development of tau PET imaging, using tracers such as [18F] flortaucipir (AV-1451), has enabled the visualization of tau tangles in the brain, a hallmark of AD pathology. Tau PET imaging has demonstrated its utility in identifying AD and distinguishing it from other neurodegenerative disorders. The visual interpretation of tau PET scans, utilizing a regional uptake scoring system, has been proposed as a method for assessing tau deposition in the brain [52,53]. Amyloid PET imaging, employing tracers like florbetapir (AV45), measures the accumulation of Aβ plaques in the brain, another key feature of AD pathology. Studies have demonstrated that amyloid PET imaging can accurately diagnose AD and monitor disease progression [50,51]. As Alzheimer’s disease progresses from MCI to moderate dementia, structural markers are more sensitive to change than Aβ deposition markers. However, in the earlier stages, from asymptomatic to MCI, amyloid markers appear to show more notable abnormalities than structural changes [43,54,55,56]. Integrating PET with other imaging modalities, such as MRI, can enhance diagnostic accuracy and provide a more comprehensive understanding of AD pathology. Multimodal image fusion methods can combine structural and metabolic information from MRI and PET scans to improve diagnosis and monitoring of AD [52,57]. Table 1 overviews the advantages and disadvantages of each neuroimaging model described.

The Standardized Centralized Alzheimer’s & related dementias Neuroimaging (SCAN) project aims to harmonize the processing of PET and MRI scans from AD research centers across the USA. This harmonization will facilitate the integration of imaging data with other ADRC data streams, resulting in a large, potentially representative cohort for research [58].

The future of PET imaging in AD research involves the development of more accurate and cost-effective PET tracers, as well as the integration of PET imaging with other biomarkers and imaging modalities. Furthermore, the application of DL and GANs to PET imaging holds promising potential for improving diagnostic accuracy and reducing the need for expensive and radioactive PET scans [59,60]. In clinical trials, AI optimizes participant selection, predicting those likely to progress to AD, thus addressing high screen failure rates and speeding up trials [3]. DL and generative adversarial networks (GANs) have been applied to PET imaging in AD research. These techniques can be used to synthesize missing PET data from MRI, improving diagnostic accuracy and reducing the need for expensive and radioactive PET scans [61,62].

## 5. XAI

XAI aims to enhance the interpretability of AI systems, making their decision-making processes more transparent and comprehensible to humans, enabling users to comprehend the reasoning behind AI-based decisions [63,64]. This is particularly important in various applications such as medical imaging [65] and healthcare [66]. XAI enhances interpretability in medical imaging and healthcare, helping clinicians understand AI-driven diagnoses and improving trust in clinical decision support [65,66]. For instance, by employing explainable ML techniques, a study aimed to elucidate the influence and correlation of features. The focus was not on creating the most accurate model but on understanding the significance and impact of each clinical measurement on diagnosis [12]. XAI would be beneficial for analyzing genetic [67], neuroimaging, and electroencephalographic data [68,69] in aid of AD biomarker detection, patient classification, and prognosis evaluation.

XAI techniques can be categorized based on their explanation scope as global and local. Local explanations address why a model made a specific decision for a single instance, highlighting the contribution of individual features, pixels, or data segments to that outcome. In contrast, global explanations describe the model’s overall logic or behavior across the entire input space, offering insights into how various factors generally influence predictions [70]. Furthermore, techniques can be divided by their timing/design into ante hoc (inherently interpretable) and post hoc (added after training a black-box model) explainability methods. Ante hoc methods—also referred to as model-based explanations—are intrinsically transparent by design, often using simpler algorithms such as linear or logistic regression, decision trees, k-nearest neighbors, fuzzy inference systems, rule-based learners, general additive models, or Bayesian models [71,72,73]. While these approaches can be very intuitive, they may lose explainability, transparency, or interpretability; particularly, in high-dimensional settings with intricate interaction effects or deep decision trees, interpretation can become challenging [74].

Conversely, machine learning algorithms such as random forests, support vector machines, and neural networks (including deep neural networks) are, within practical constraints, inherently non-explainable and are typically referred to as “black-box” models [70,72,75]. In these cases, post hoc methods—often model-agnostic—are applied once the model is fully trained, without requiring internal access to its parameters. Rather than dissecting the model’s architecture, these post hoc techniques generate insights by examining the input–output behavior (for instance, through perturbations or derivative-based analyses) and can yield either local or global explanations depending on the application [72,73,76].

A common perception exists regarding a trade-off between a model’s performance (predictive accuracy) and its explainability [70,72,73]. High-performing algorithms, such as deep learning, are often the least explainable [73]. As a result, simpler models are often preferred for their interpretability, even though there is a common belief that improving explainability may come at the cost of model performance [77,78]. However, linear models are not necessarily more interpretable than neural networks, particularly when high-dimensional or heavily engineered features are employed, which can diminish the model’s interpretability or explainability [76]. Furthermore, more complex models do not always guarantee higher accuracy [79].

Post hoc XAI methods are commonly classified into two main categories: gradient-based approaches and occlusion-based techniques [80]. Gradient-based techniques, a visualization approach in XAI, compute the derivative of a model’s output concerning each input feature, such as individual pixels in an image, to highlight influential regions. A large gradient suggests that even a minor alteration in that dimension can significantly impact the model’s output, indicating the dimension’s importance. Unlike gradient-based methods, occlusion-based techniques do not depend on gradients to interpret a model’s behavior. This allows them to be applied to models with minimal or flat gradients, as well as non-differentiable models. The principle of occlusion-based methods is simple: each input dimension (such as a pixel or a specific region in an image) is systematically masked to assess its impact on the model’s predictions. If the change is substantial, the altered dimensions are likely important to the model. By monitoring these changes, a heatmap can be generated, indicating which patches in the image are more or less critical to the model’s predictions [80].

Additionally, techniques can be classified by their applicability, either as model-agnostic or model-specific [70]. Model-specific explanation methods are constrained to particular classes of models. For instance, such a method might utilize attributes that are unique to a specific type of neural network. A limitation of this approach is that by focusing on model-specific explanations, the range of applicable neural networks is restricted, potentially excluding a neural network that could more effectively fit the output to the input data. In contrast, model-agnostic methods operate purely on the model’s inputs and outputs, making them inherently post hoc. Because they do not rely on a model’s inner workings, model-agnostic approaches can be applied to any black-box model, enhancing flexibility and transferability across different architectures [65].

## 6. XAI in Neuroimaging

XAI plays a pivotal role in the development of AI models for AD diagnosis. Also, XAI methods help identify the most relevant features and biomarkers that contribute to AD diagnosis, such as white matter hyperintensities (WMHs), which XAI techniques have revealed as significant biomarkers used by DL models for AD identification [81], and brain atrophy patterns captured by Graph Convolutional Networks (GCNs) that predict cognitive status ranging from normal cognition to MCI and AD [82]. Moreover, XAI approaches have been employed to predict the conversion from MCI to AD. Various approaches and frameworks have been proposed for XAI in AD diagnosis, including LIME, SHAP, Gradient-weighted Class Activation Mapping (GradCAM), and Layer-wise Relevance Propagation (LRP). As depicted in Table 2, we summarized the studied tasks and obtained accuracy range in various XAI models discussed in AD neuroimaging.

### 6.1. LIME (Local Interpretable Model-Agnostic Explanations)

LIME is a post hoc technique designed to provide local explanations of a single prediction regardless of the underlying model architecture. It achieves model agnosticism by perturbing the input around the instance of interest and observing how these variations affect the model’s output [83,84]. In the context of image data, LIME typically works at the superpixel level (conceptually similar to an occlusion or masking approach) [83]. This concept of local fidelity emphasizes that features that are important for a single sample may not be significant globally [85]. Moreover, because LIME relies on random sampling, it can cause variability, meaning different runs on the same data may not always yield identical explanations [84]. LIME simplifies a complex model (e.g., deep neural networks or complex decision trees) by fitting a local linear approximation. This means that if the original model relies on nonlinear interactions between features, those interactions might be lost in the LIME explanation, making it less accurate in capturing the true decision process of the model [86]. It has been effectively applied in several areas, including brain tumor detection through MRI images, where the Deep Explainable Brain Tumor Deep Network (DeepEBTDNet) model utilized LIME to provide clear explanations for its high-accuracy predictions [87]. Additionally, it has been utilized in cardiomyocyte beating analysis [88], disease classification, and rapid diagnosis of COVID-19 [89].

In the field of AD and neuroimaging [19,24,90,91,92], Duamwan et al. used LIME with CNNs to detect AD from MRI scans. LIME was used to provide visual proof by highlighting image regions that contributed to predictions [91]. In other investigation, Kamal et al. applied LIME to gene expression data, ranking genes based on their probability of indicating AD [90]. Shad et al. explored neural network models for early AD detection using a hybrid dataset from Kaggle and OASIS [93]. Khan et al. [32] proposed a fractional order-based CNN classifier with XAI for classification of AD diagnosis. Furthermore, Adarsh et al. [33] introduced an innovative diagnostic framework, combining CNNs with Multi-feature Kernel Supervised within-class-similar Discriminative Dictionary Learning (MKSCDDL) to classify AD, MCI, and CNs.

### 6.2. SHAP (SHapley Additive exPlanations)

SHAP is a post hoc, model-agnostic XAI method designed to quantify the contribution of each feature to a model’s prediction [94]. SHAP works by using Shapley values, a concept from game theory, to measure how much each feature contributes to the model’s prediction. It achieves this by evaluating the impact of each feature in all possible combinations with other features [85]. Although this combinatorial approach offers local and potentially global insights, it is also computationally intensive; exact computation quickly becomes intractable for deep neural networks or high-dimensional data. To address this, model-specific implementations exist for tree-based methods (e.g., random forests, XGBoost), which can reduce the computational burden [84].

SHAP has several pitfalls that require careful consideration. One major issue is the misinterpretation of its scores, as SHAP values indicate feature ranking rather than direct causal influence on the outcome. Users should focus on the order of features rather than assuming that higher scores correspond to stronger effects. Additionally, SHAP is susceptible to biases within the model, meaning it can produce misleading explanations if the classifier itself is biased. Another limitation stems from its assumption that features are independent, which is often unrealistic in real-world datasets where variables can be highly correlated. This assumption can lead to misallocated importance scores, where significant features receive lower scores simply because their effect is captured by other correlated variables [86].

Another strength of SHAP is its flexibility: it can handle tabular, text, and image inputs by mapping them into Shapley value computations, making it a proper choice for explaining a wide range of ML applications [94]. It is widely applied in medical imaging to explain CNN-based predictions for renal anomalies in CT scans, histopathology classification [95], and brain tumor detection in MRI images [96,97]. In radiomics, SHAP enhances interpretability in disease predictions like Radiation Pneumonitis (RP) [98] and thyroid cancer metastasis assessment [99].

In the research field related to AD, a study by El-sappagh et al. [100] aimed to provide a comprehensive understanding of how various modalities influence the risk of AD. The study incorporated 11 modalities from the ADNI dataset and employed a random forest (RF) classifier. The model delivers precise diagnoses and elucidates each decision using the SHAP framework. In another study, Bogdanovic et al. [12] analyzed a large dataset of medical, cognitive, and lifestyle measurements.

Feature selection and disease progression modeling have also benefited from SHAP. Bloch et al. [101] used Data Shapley to refine patient selection in both random forest and XGBoost models. Lombardi et al. [102] illustrated how SHAP values can show the influence of each cognitive index on a patient’s cognitive status and offers valuable insights into the progression of AD.

The TADPOLE Challenge, a comprehensive competition for predicting AD progression, identified tree-based ensemble methods as the most effective for clinical status prognosis. Hernandez et al. [14] used SHAP to identify key features in a Random Forest model, showing that accuracy remained stable with 1818 features but declined when essential cognitive features were removed. Chun et al. [103] applied SHAP for both local and global interpretation in predicting the transition from MCI to AD, ensuring a more explainable decision-making process. Jahan et al. [15] developed a multimodal approach that combined data from multiple datasets, with SHAP identifying key features such as judgment, memory, and orientation as the most critical for diagnosis. Another investigation executed a threefold classification among control subjects, cognitively impaired individuals, and dementia patients, using SHAP to elucidate RF classifier decisions, employing a leave-one-subject-out cross-validation strategy [102]. Zhang et al. [104] introduced a patch-based convolutional network for AD diagnosis, where SHAP was used to identify the most significant image patches in a transfer learning model, reducing computational complexity by optimizing the number of patches required for classification. Xu et al. [105] proposed a multiclass classification model that incorporated SHAP for both single-instance and global interpretations. Ekuma et al. [106] compared multiple CNN models, using SHAP to quantify the influence of specific brain regions on AD classification. Graph-based approaches and biomarker analysis have also benefited from SHAP’s explanatory capabilities. Amoroso et al. [10] explored the combination of graph theory models with Shapley values to distinguish between clinical conditions using T1 brain MRI data. Their findings highlighted significant brain regions, such as the putamen, middle and superior temporal gyrus, hippocampus, amygdala, posterior cingulate, and precuneus, that play a significant role in AD and MCI.

Additional studies by Yilmaz et al. [107] and Rahim et al. [108] utilized SHAP to improve the interpretability of deep learning models for AD severity prediction. Yi et al. [109] addressed class imbalance in AD diagnosis using an XGBoost-SHAP framework, providing valuable clinical decision-making insights. Furthermore, Bhattarai et al. [27] launched the Deep-SHAP method, which integrates deep learning with SHAP, to map relationships between regional neuroimaging biomarkers and cognition in MCI/AD patients. The method detected crucial brain regions, like the insula and temporal pole, involved in cognitive impairment and demonstrates the potential of AI-based diagnostic tools. In a cross-sectional study, Leandrou et al. [11] employed an XAI algorithm to classify AD utilizing MR radiomics. They employed radiomic features from the entorhinal cortex and hippocampus, combined with neuropsychological evaluations, to train an XGBoost model, explained with SHAP values. Sarica et al. [17] assessed the sex disparities in the conversion risk from MCI to AD using Random Survival Forests and SHAP. They identified that male models rely more on CSF and neuroimaging features, while female models depend more on neuropsychological tests. De Francesco et al. [30] developed and interpreted the MUQUBIA algorithm to distinguish various dementias using MRI and clinical data. By using SHAP, MUQUBIA can categorize dementia with high precision (88%), benefiting from cost-effective clinical and MRI information. Khan et al. [31] introduced a genetic programming (GP) model for MRI image enhancement and an XAI framework to predict AD severity. The framework applies CNN models and oversampling methods to address data imbalance, enhancing the interpretability of predictions from MR images. Rashmi et al. conducted a comprehensive study utilizing various MRI feature datasets to diagnose AD with a SHAP XAI model. Brusini et al. [24] used XAI to emphasize the most significant characteristics in SVM classifications and to validate the results by seeing the explanations’ repetition across different techniques. The explainability analysis indicated the central role of the cingulum in showing early signs of AD. Vetrithangam et al. [19] fostered an interpretable deep learning model for detecting AD, incorporating XAI models such as Grad-CAM, SHAP, and LIME. The study employed an Enhanced Fuzzy C-Means algorithm to refine feature classification and enhance the interpretability of the model’s decision-making process.

However, challenges such as handling high-dimensional data, scaling SHAP to accommodate larger datasets and more complex models, and integrating SHAP with other XAI methodologies like LIME and Grad-CAM to provide a more holistic understanding of the model’s decision-making processes remain. Addressing these challenges is crucial for fully employing SHAP’s potential in imaging applications, thus underscoring its pivotal role in the field of XAI.

### 6.3. LRP (Layer-Wise Relevance Propagation)

LRP is a post hoc model-specific method which was originally applied to deep learning models. It works by backpropagating and starts from the model’s output and redistributes “relevance scores” layer by layer until it reaches the input layer, highlighting which parts of the input contributed the most to the prediction. Because of this approach, it has low computational costs and, in contrast to heuristically based models (like SHAP and LIME), uses exact mathematical derivation to make decisions [84]. LRP generates heatmaps for Alzheimer’s detection in MRI [110], aids 3D CNNs in classifying preterm births [111], offers pixel-level insights for blood cancer diagnosis [112].

Building on LRP, Pohl et al. [113] introduced composite LRP, which employs multiple propagation rules instead of a single rule. Their study quantitatively compared interpretation measures between uniform LRP and composite LRP, revealing that composite LRP provides more focused visualizations of relevant brain regions for positive AD classification. This technique effectively filters out less relevant areas, offering clearer relevance heatmaps. In another paper by Sudar et al. [114], they employed LRP to identify the stages of AD using brain images, alongside other algorithms like VGG-16. In their investigation, Leonardsen et al. [115] aimed to find neuropathological aspects of the disease by means of XAI. In this paper, CNN was trained to distinguish between dementia patients and healthy controls, and LRP was employed to offer explanations of the model predictions at the individual level. The study showed that the patterns identified by the AI model were consistent with what is now understood about the neuropathology of dementia. Böhle et al.’s [110] investigation into LRP demonstrated how this method not only visualizes CNN decisions but also identifies critical regions within MRI scans by summing all layer-wise relevance metrics.

### 6.4. SMs (Saliency Maps)

SMs are post hoc model-specific tools which work by computing the gradient of the model’s output with respect to the input image, where higher gradient values indicate pixels that have the greatest impact on the decision. The intensity of each pixel reflects its saliency, showing how much it contributes to the model’s classification. These maps, often called heatmaps, highlight the most influential areas in an image.

In their study, AbdelAziz et al. [26] proposed an advanced interpretable diagnosis of AD using the SECNN-RF framework. This study implemented SM to obtain visual explanations of the model’s decisions. Additionally, De Santi et al. [13] discussed the development of an explainable CNN for early AD diagnosis using 18F-FDG PET images. This model performed multiclass classification and used post hoc explanation techniques like SMs and LRP.

### 6.5. GradCAM (Gradient-Weighted Class Activation Mapping)

GradCAM is a model-agnostic tool that can applied to different ML models. Grad-CAM highlights the regions in an image that most influence a model’s decision using gradient information from the last convolutional layer. GradCAM mostly depends on the last layer, making it faster and computationally cheap. However, it may not always capture small details or multiple objects well [116].

Grad-CAM has been utilized in various medical imaging applications, including the classification of multiple sclerosis subtypes using clinical brain MRI data [117], detecting AD from MRI scans [118], diagnosis of COVID-19 from chest X-ray images [119], and brain tumor detection [120].

One notable application is the 3D Residual Self-Attention Deep Neural Network (3D ResAttNet), which utilizes sMRI data to enhance AD diagnosis accuracy. Zhang et al. employed GradCAM with a 3D ResAttNet to produce heatmaps that highlighted features from MRI scans for each network layer. GradCAM’s ability to work with multimodal inputs without requiring architectural changes or retraining makes it a valuable tool for enhancing the interpretability of deep learning models [16]. In another study, Ruengchaijatuporn et al. used images from bedside tasks, such as clock drawing tests and cube-copying, to classify healthy controls and AD patients using a deep neural network. Additionally, it provided superior interpretability scores and IoUs compared to a multi-input model utilizing Grad-CAM visualization. Their work also compared CNN outputs with those from VGG16, providing visual explanations through GradCAM [121]. Jain et al. constructed heatmaps from MRI scans using GradCAM for each CNN layer and combined them for final interpretation. While Grad-CAM primarily offers qualitative visualizations, it also contributes to quantitative analysis by evaluating model attention and localization, enhancing trust, particularly in healthcare applications [122].

Furthermore, Mahmud et al. [22] suggested an XAI-based approach for AD detection using deep transfer learning and ensemble modeling. It uses SMs and grad-CAM to improve interpretability, providing visual insights into neural regions influencing the diagnosis. In their study, Song et al. [18] utilized Grad-CAM with an fMRI-based 3D-VGG16 network for AD diagnosis, aiming to identify key brain regions of interest (ROIs) that the model prioritizes during prediction. Coluzzi et al. [20] compared a DL model using structural connectivity with an MRI model, using a novel XAI approach based on Grad-CAM. It quantitatively evaluates the performance of these models in comparison to established AD biomarkers, assessing their effectiveness in the decision-making process. In addition, Bapat et al. [28] conducted experiments using raw data from the ADNI dataset to predict conversion to AD, achieving an accuracy of 0.834. They employed Grad-CAM to enhance interpretability, highlighting regions such as the putamen, thalamus, amygdala, frontal pole, frontal gyrus, and planum polare.

### 6.6. LEAR (Learn–Explain–Reinforce)

The Learn–Explain–Reinforce (LEAR) framework integrates the processes of learning a diagnostic model, reinforcing it, and generating visual explanations. This system aims to identify potential irregularities in brain imaging that could lead to AD diagnosis. The use of counterfactual maps within this framework, validated using the ADNI dataset, demonstrates its integrity and comprehensibility [123].

### 6.7. GNNExplainer (Graph Neural Networks Explainer)

GNNExplainer, or GNNE, is a model-specific method designed for Graph Neural Networks (GNNs), providing insights into model decision-making by analyzing subgraph structures and node features. It works by maximizing the mutual information between a model’s prediction and the distribution of possible subgraphs, highlighting node and feature importance in the process. Node importance is determined using degree centrality (DC), which quantifies the influence of a node within the subgraph, while feature importance is assessed using a feature node mask, which pinpoints the most relevant attributes influencing the model’s decision. Additionally, GNNE can provide comprehensive global explanations for every type of instance [124]. By successfully combining longitudinal neuroimaging and biologically significant data, Kim et al. [124] presented an interpretable GNN model for AD prediction. They employ GNNE to pinpoint important nodes that play a role in the prediction, forming a subgraph structure along with a selection of node attributes essential for the forecasting process. Two primary advantages that make GNNE a useful tool is its ability to reveal syntactically relevant structures and interpretations and the potential to understand errors in GNNs. In another study, Kim investigated GCNs to provide accurate predictions along with interpretable results, thereby contributing to a more comprehensive understanding of individual AD prognoses. The research utilized GNNExplainer, which optimized a subgraph within an individual’s neighborhood and identified essential features critical to the prediction [29].

### 6.8. Occlusion Sensitivity Method

Occlusion Sensitivity is a model-agnostic, local explanation method that systematically hides parts of an input to measure the effect on a model’s prediction. By masking regions (e.g., pixels in an image, words in a text, or features in tabular data) and observing output changes, it identifies important features. If occluding a region significantly alters the prediction, that region is deemed important. This method does not require access to the model’s internals, making it applicable to any black-box model. It is primarily used for individual-instance explanations rather than global model behavior [125].

Bordin et al. employ the Occlusion Sensitivity method to reveal the relevance of white matter hyperintensity lesions in comparison to healthy lesions. This technique involves removing a patch from the input dimension of an image and comparing the output to determine the image’s susceptibility to occlusion in various regions. A significant variation upon patch removal indicates its importance for classification. The authors successfully classified the brain areas contributing most to the classification using the Occlusion Sensitivity technique. This study offers insights into the image attributes utilized by the network for specific classifications and highlights potential misclassifications [81]. Similarly, Rieke et al. use the Occlusion Sensitivity method to visualize heatmaps that distinguish between healthy controls and AD [126].

### 6.9. Three Different Approaches

Upon encountering studies that utilized less-well-established XAI frameworks, explaining their approaches in greater detail in this section is necessary. García-Gutiérrez et al. [21] analyzed data from 1617 participants in the ADNI dataset. The researchers employed [18F]-fluorodeoxyglucose positron emission tomography (FDG-PET) to assess brain metabolism. The study identified brain regions exhibiting significant hypometabolism in AD and leveraged deep learning models to predict future metabolic changes. Their findings revealed a strong correlation between hypometabolism, AD progression, and cognitive decline. To enhance interpretability, the Integrated Gradients (IG) method was applied to the multi-task model, highlighting the posterior cingulate cortex (PCC) and left putamen as the most critical regions. In a study, Cai et al. [23] used an Explainable Boosting Machine (EBM) that integrated multimodal features to forecast the transition from MCI to AD across different follow-up periods. Data for this analysis were drawn from the ADNI database, encompassing records of 1042 MCI patients collected between 2006 and 2022. The exposures included MRI biomarkers, cognitive test scores, demographics, and clinical features. The five-year prediction accuracy reached 85% (AUC = 0.92) using both cognitive scores and MRI markers. In the early stage of AD, MRI markers played a significant role, while cognitive scores were more important for the mid-term. Lower right inferior temporal volume, low left inferior temporal thickness, higher FAQ scores, APOE4, and lower right entorhinal volume were identified as critical features for one-, two-, three-, four-, and five-year predictions, respectively. The EBM model provided global and local explanations. Pairwise interactions were also considered in the model. Furthermore, Saad Saoud and AlMarzouqi [25] proposed a method for predicting AD and MCI using MRI data, combining an ROI-based approach with deep learning in an interpretable framework. They applied three-dimensional vision transformers (3D-ViTs) to analyze each ROI individually, while a Deep Belief Network (DBN) served as an ensemble learning model to enhance predictive accuracy. The approach was evaluated on the baseline structural MRI dataset obtained from the ADNI cohort. The proposed system outperformed existing models, achieving an accuracy of 90% for the AD vs. CN task and 84% for the CN vs. MCI task. The study enhanced explainability by segmenting MRI brain scans into 138 distinct volumes corresponding to clinically relevant subregions, enabling a detailed examination of regional brain abnormalities.

## 7. Datasets for Training XAI Models in AD Neuroimaging: Accessibility and Challenges

A key factor in developing reliable AI models, especially in neuroimaging, is the quality and quantity of data used for training. AI models, including those that incorporate XAI techniques, can only be as good as the data they learn from [127].

Publicly available medical imaging datasets including the Alzheimer’s Disease Neuroimaging Initiative (ADNI), Open Access Series of Imaging Studies (OASIS), and BrainLat are invaluable for advancing XAI in AD neuroimaging research [128,129].

ADNI offers comprehensive longitudinal data, including neuroimaging modalities, genetic data, cerebrospinal fluid biomarkers, and cognitive assessments (including ADAS-Cog, MMSE, and MoCA), accessible under a Data Use Agreement. These data are obtained from over 2500 participants and constitute diverse imaging scans like FBB-PET, FBP-PET, FTP-PET, and MPRAGE T1-weighted MRI [130,131].

OASIS provides more open access to MRI and clinical data, facilitating easier adoption for research purposes [132]. OASIS-3 features data from 1378 participants with 2842 MR sessions (including T1w, T2w, FLAIR, SWI, resting-state BOLD, and DTI sequences) and 2157 raw PET scans, along with cognitive assessments from the Uniform Data Set (UDS) [133].

While BrainLat contributes multimodal neuroimaging and cognitive data, access may be limited due to specific institutional agreements. BrainLat collects data from 780 participants, offering anatomical MRI, resting-state fMRI, diffusion-weighted MRI (DWI), and EEG, supplemented by cognitive tests like Montreal Cognitive Assessment (MoCA), INECO Frontal Screening (IFS), and Facial Emotion Recognition (FER).

The accessibility and richness of ADNI and OASIS make them particularly suitable for XAI research, helping to improve our understanding and treatment of AD.

Although these datasets contribute significantly to advanced AI model training, they encounter several challenges. One of the biggest issues is the balance between dataset size and data quality. Large datasets help models generalize better; poor-quality or inconsistent labels can reduce accuracy and trustworthiness [134]. Annotation quality is another major concern. Since neuroimaging data are typically labeled by experts, differences in interpretation among radiologists or neurologists can lead to inconsistencies. Establishing standardized annotation guidelines and ensuring agreement among experts can help improve reliability. Another challenge is dataset bias and many available datasets lack complete labels for all disease stages, which can skew model predictions [135]. Additionally, variations in imaging protocols, scanner resolutions, and patient demographics make it difficult for models to perform equally well across different clinical settings. To address these issues, researchers are exploring domain adaptation techniques and other strategies to ensure AI models are not only accurate but also adaptable and clinically meaningful. Tackling these challenges is essential for making XAI-based neuroimaging models both effective and trustworthy in real-world applications.

## 8. Discussion

As overviewed in Figure 2 and Table 3 in the present investigation, we conducted a review of the advantages associated with the application of XAI methodologies within the domain of neuroimaging, in the context of AD. Various methods of XAI have been employed in different studies, such as LIME, SHAP, LRP, SMs, GradCAM, GNNE, and Occlusion Sensitivity, which you can see the distributions of in Figure 3. Each of these techniques possesses its own unique strengths and achieves varying levels of accuracy.

Choosing the best XAI method depends on the data modality, the need for local or global insights, and their applicability, computational resources, and required speed. These varieties necessitate the development of an algorithm to select the desired algorithm to implement accurately. Figure 4 depicts a model for the required diagram.

Regarding the suitability of XAI models for different data types, LIME, LRP, and SHAP are used for tabular and mixed (e.g., clinical and genomic) data such as text and image data. Grad-CAM and Occlusion Sensitivity, in contrast, are primarily adopted in imaging contexts—particularly for CNN-based models and MRI or CT scans—where they generate heatmaps highlighting important regions [94,136]. GNNE (Graph Neural Network Explainer) is specialized for graph-structured data such as brain connectivity networks [124].

In terms of local versus global explanations, LIME, Grad-CAM, LRP, SM, GNNE, and Occlusion Sensitivity are predominantly local methods focused on explaining individual predictions. SHAP can used both locally and in an aggregated manner to derive global insights.

Speed varies among these methods. Grad-CAM and LRP tend to be relatively fast once the network is trained, as they involve additional gradient or relevance backpropagation steps. Occlusion Sensitivity can become computationally expensive for high-resolution 3D images, as each image region must be systematically masked to observe how the prediction changes. LIME and SHAP can also be slow for large datasets or models: LIME must repeatedly train local surrogate models around a chosen instance, while SHAP (especially its exact form) has high computational demands. Consequently, Grad-CAM or LRP can be more practical in near-real-time diagnostic settings, whereas LIME and especially SHAP might be more suitable for offline or post hoc analyses rather than immediate bedside decisions [137,138].

**Figure 2 diagnostics-15-00612-f002:**
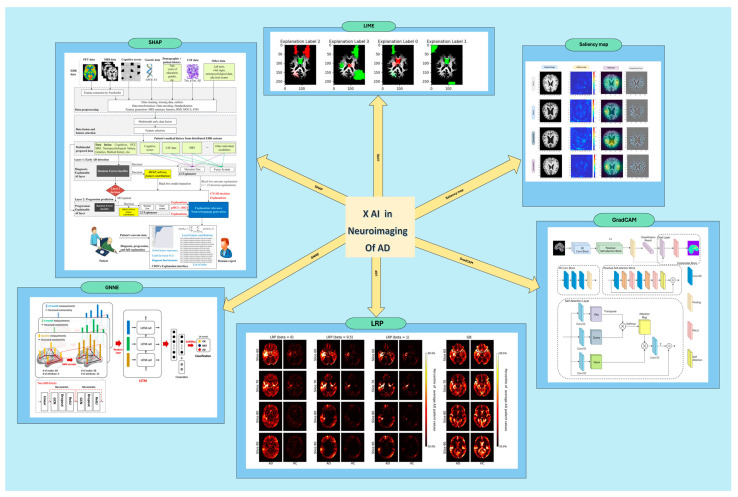
This figure illustrates various XAI methods applied to neuroimaging data for AD. SHAP: The XAI framework integrates multiple data modalities to build robust predictive models and provides explanations for the overall behavior of RF models and individual prediction outcomes using the SHAP method. LIME: The figure includes a visual representation of the LIME output, showing how specific predictions are influenced by different code sections. LRP: Additionally, the figure presents average heatmaps for all subjects in the test set, plotted separately for LRP with β = 0 on the left and Guided Backpropagation (GB) on the right. GNNE: The image also highlights an interpretable GNN model designed for prognosis prediction, emphasizing its structure and interpretability features. Saliency map: The column next to the original brain MRIs presents the saliency map results for pre-trained models of VGG-16, VGG-19, DenseNet-169, and DenseNet201, showcasing the accurate localization of AD-affected areas of the patient’s brain. GradCAM: Finally, the figure showcases a deep neural network architecture that utilizes 3D residual connections and attention (ResAttNet34) mechanisms, on which the 3D GradCAM was applied, enhancing the model’s ability to discern complex spatial patterns and improving accuracy and performance (reprinted with CC BY from [22,89,100,124,139]).

**Figure 3 diagnostics-15-00612-f003:**
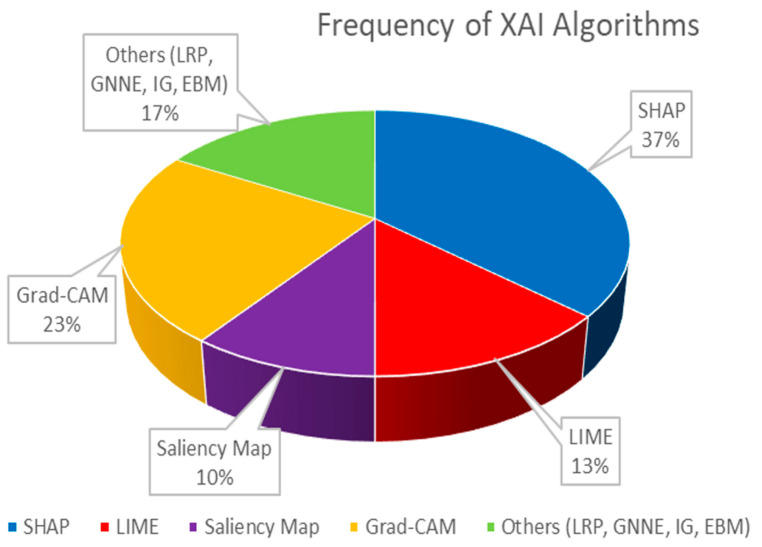
Distribution of XAIs in the scope.

SHAP provides more consistent explanations than LIME due to its game-theoretic foundation [94]. However, unlike heuristically based approaches like SHAP and LIME, LRP relies on exact mathematical derivations to decompose decision-making, making it the most faithful explanation method [84].

From a resource-cost perspective, once the model is trained, Grad-CAM and LRP demand minimal extra information, making them relatively feasible. LIME and SHAP, particularly with large data or models, require significant computational resources, thus often restricting their usage to offline analyses. Occlusion Sensitivity also becomes expensive for 3D imaging if systematically checking numerous patches [116,137].

**Figure 4 diagnostics-15-00612-f004:**
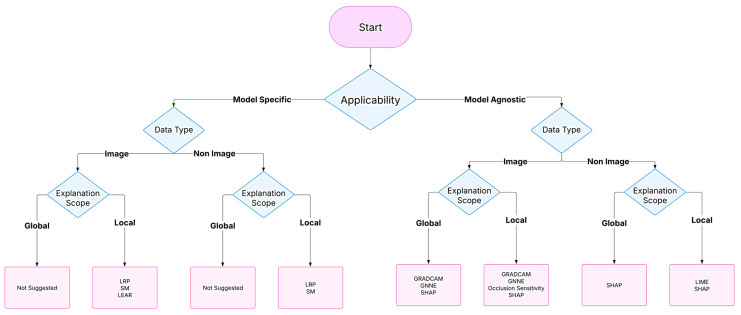
The algorithm overviews the suggested utilization of XAI methods based on the study’s aim and model applicability, data type, and explanation scope.

The utilization of diverse visualization techniques is essential for gaining insights into the decision-making processes of CNNs. This is a critical step towards enhancing the clinical relevance and trustworthiness of computer-based decision support systems [126]. Jain et al. utilized Grad-CAM as the “Model Visualization” stage. The third phase of the process entailed the implementation of Grad-CAM-based feature visualization. This involved extracting the weights and features from the final convolutional layer and subjecting them to visualization using Grad-CAM [122].

Each study focused on different aspects of the disease. For instance, the primary aim of a study by Lombardi et al. [102] was to investigate whether the entire vector of feature importance generated by XAI could serve as a novel cognitive marker across the spectrum of cognitive abilities, rather than relying on a collection of original features. Their results indicate that SHAP values can effectively capture the influence of individual indices on patients’ cognitive status and measure the fluctuations in this influence over time. This approach tracks longitudinal changes and offers continuous insights into the progression towards AD. Chun et al. [103], in their research, offer a local explanation for the prediction of conversion from amnestic MCI to dementia or AD using Individual Conditional Expectation [58] and SHAP for each patient. Their study demonstrated that the XGBoost algorithm achieves the best prediction performance. ICE plots illustrate the variation in a feature of interest while maintaining the values of other features constant, thereby providing insights into how each individual instance responds to changes in a specific feature [103]. In other study, Jahan et al. argued that utilizing multimodal data showed promising results. Furthermore, to ensure the reliability of this model’s predictions, the SHAP explainer is employed, which reveals all decision-making features along with their respective values. The results from the explainer indicate that Judgment, Memory, Homehobb, Orient, and Sumbox are the most significant features [15].

The available imaging modalities each have their own strengths and limitations in the context of AD diagnosis and assessment. sMRI provides widespread access to safe scanning techniques and can correlate brain volume changes with cognitive decline, but it lacks specificity in identifying reduced hippocampal volume [42] and struggles to directly visualize NFTs and Aβ plaques, as well as encountering challenges in detecting atypical AD cases [140]. fMRI offers a non-invasive assessment of brain function, including the ability to correlate abnormal activity with areas of amyloid deposition and explore functional connectivity between brain networks, yet it is unable to detect Aβ plaques and NFTs [141], is sensitive to head motion [142], is costly, and has restricted temporal resolution. FDG-PET demonstrates high sensitivity and specificity in predicting AD conversion risk and providing early identification of reduced glucose metabolism, and can aid in mapping the topographic progression of the disease, but faces challenges related to availability, cost, invasiveness from radiotracer injection, and non-specific glucose hypometabolism in AD [143]. Amyloid-PET imaging allows for the visualization of Aβ plaques, a key feature in early AD diagnosis and the prediction of MCI conversion to AD, yet it faces a weak association between Aβ deposition and disease severity, invasiveness from radiotracer injection, variability in uptake among healthy individuals, and limited understanding of Aβ accumulation in atypical AD presentations [144]. Finally, Tau-PET Imaging provides a strong link between tau accumulation and cognitive decline, high affinity of radiotracers for pathological tau, and a robust correlation between neurodegeneration and NFTs, but is also invasive, faces variability in tau morphology across AD subtypes, and remains an evolving area of research and clinical practice [144].

## 9. Limitations, Challenges, Needs, Prospects, and Future Direction of XAI in AD Neuroimaging

XAI in neuroimaging has demonstrated substantial promise in augmenting the comprehension and diagnosis of AD. However, there are several limitations, challenges, and necessities that must be addressed to fully realize the potential of XAI in this domain.

The ethical, regulatory, and privacy challenges of XAI in medical diagnostics are as crucial as its technical advancements. Trust and accountability are key concerns, as AI explanations must be clear and reliable for both clinicians and patients to prevent misdiagnosis and over-reliance on automated decisions. Bias in AI models, stemming from imbalanced datasets, must be addressed to ensure fairness in predictions. Regulatory hurdles, including compliance with FDA, EMA, and MHRA guidelines, require rigorous validation before AI tools can be deployed in healthcare [145]. Privacy concerns, particularly in neuroimaging, demand adherence to HIPAA and GDPR regulations, with solutions like data de-identification and federated learning helping to protect sensitive patient data [146]. Overcoming these challenges is essential for fostering trust and ensuring that XAI can be safely and effectively integrated into Alzheimer’s disease diagnosis and broader medical applications.

XAI models necessitate expansive, high-quality datasets to learn and generalize effectively. Nonetheless, the availability of such datasets is constrained, particularly for neuroimaging data, which can be costly and time-consuming to amass [147,148,149]. While XAI aims to provide insights into the decision-making process, the interpretability of these models can be difficult, especially for complex neuroimaging data. This lack of transparency can engender skepticism among healthcare professionals and the general public [148,150,151]. Many neuroimaging datasets for AD diagnosis are imbalanced, with a substantial proportion of healthy controls and a smaller number of AD cases. This imbalance can adversely impact the performance of XAI models and require additional techniques for mitigation [150]. Neuroimaging data can be computationally intensive, necessitating significant resources for processing and analysis. This can pose a challenge for researchers and clinicians who may lack access to such resources [152]. The utilization of diverse neuroimaging modalities, acquisition protocols, and preprocessing techniques can lead to variability in results. Standardization and reproducibility are crucial for ensuring the reliability of XAI models and their applications [152,153,154].

The development of standardized datasets and protocols for neuroimaging data collection and analysis is essential for ensuring the reproducibility and generalizability of XAI models [152,153,154]. Methodologies like transfer learning and data augmentation can aid in addressing the limitations posed by limited data availability. These techniques can be particularly beneficial for fMRI-based DL and other neuroimaging applications [148,150]. The advancement of more interpretable and transparent XAI models is pivotal for cultivating trust among healthcare professionals and the general public; techniques such as gradient class activation maps and feature importance can contribute to this objective [150,155]. Collaboration among researchers, clinicians, and industry experts is necessary for the progression of the field of XAI in neuroimaging. Knowledge sharing and the development of open-source tools can facilitate this collaboration [152].

The future direction of XAI in the neuroimaging of AD involves several key areas of research and development. The integration of various neuroimaging techniques, such as MRI, PET, and DTI, is poised to enhance diagnostic precision and provide a comprehensive understanding of diseases [156,157]. The integration of neuroimaging and genetic data through deep learning approaches will identify key features of tau protein accumulation and potential genetic markers associated with AD [158]. Advancements in neuroimaging technologies like functional MRI and diffusion tensor imaging will provide more detailed insights into brain structure and function, enabling more accurate diagnosis and monitoring of AD [159]. Finally, future research in XAI for Alzheimer’s neuroimaging should focus on systematically evaluating explainability methods beyond interpretability alone. Key aspects include faithfulness, ensuring that explanations truly reflect the model’s decision process; stability, measuring consistency under small input variations; completeness, assessing whether all relevant features are captured; and interpretability, ensuring explanations are understandable to clinicians. Human-centered evaluations should test how well XAI methods improve diagnostic decision-making, while counterfactual metrics can assess the plausibility of alternative explanations.

## 10. Conclusions

XAI plays a vital role in improving the transparency and dependability of AI-based neuroimaging tools for AD. Although deep learning models excel in detecting AD biomarkers and forecasting disease progression, their opaque decision-making process limits their clinical adoption. Various XAI techniques, including SHAP, LIME, Grad-CAM, and LRP, help clarify how these models prioritize features, enabling better clinician understanding and increasing trust in AI-driven diagnostics. However, challenges such as limited data availability, computational demands, regulatory constraints, and inconsistencies in imaging standards hinder widespread implementation. Advancing XAI for AD neuroimaging requires collaborative efforts, multimodal data integration, and the development of more interpretable models. Future research should enhance explainability by ensuring faithfulness, stability, completeness, and clinician-centered evaluations.

## Figures and Tables

**Figure 1 diagnostics-15-00612-f001:**
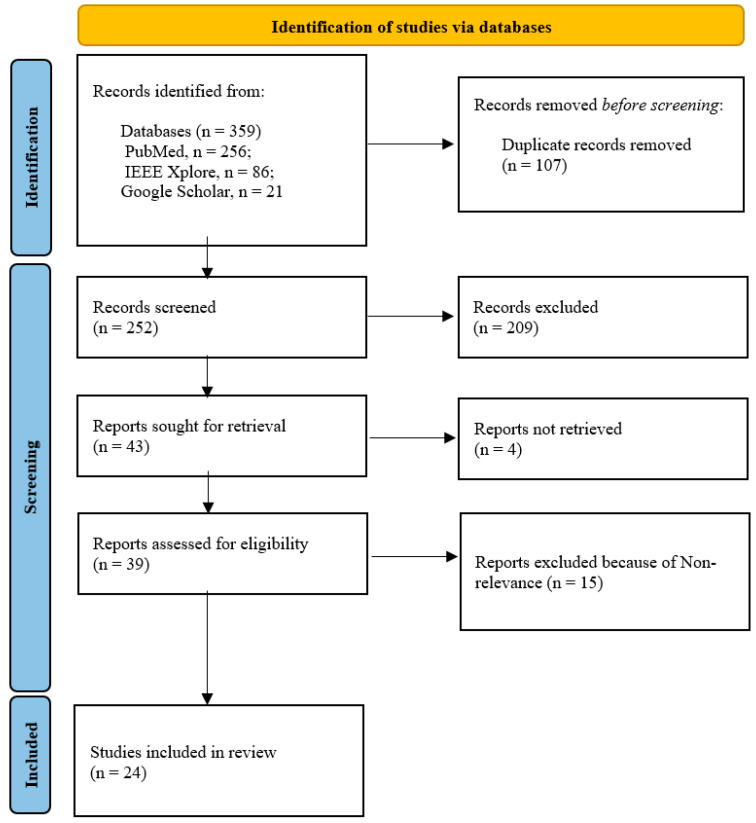
Article selection flowchart.

**Table 1 diagnostics-15-00612-t001:** Overview of pros and cons of using different neuroimaging modalities in AD.

Imaging Modality	Advantages	Disadvantages
Structural MRI (sMRI)	- Widespread availability of safe MRI scanners - Ability to predict brain volume - Correlation of atrophy with cognitive decline	- Lack of specificity in identifying reduced hippocampal volume - Inability to directly visualize NFTs and Aβ plaques - Inconsistent patterns of atrophy in different AD subtypes - Challenges in detecting atypical AD cases
Functional MRI (fMRI)	- Non-invasive assessment of brain function - Correlation of abnormal activity with areas of amyloid deposition - Exploration of functional connectivity between brain networks	- Inability to detect Aβ plaques and NFTs - Sensitivity to head motion - High cost - Restricted temporal resolution
Fluorodeoxyglucose–Positron Emission Tomography (FDG-PET)	- High sensitivity and specificity in predicting AD conversion risk - Early identification of reduced FDG uptake - Mapping topographic progression to aid in AD variant identification	- Challenges related to availability - Cost issues - Invasiveness from radiotracer injection - Non-specific glucose hypometabolism in AD
Amyloid-PET Imaging	- Visualization of Aβ plaques - Key feature in early AD diagnosis - Prediction of MCI conversion to AD	- Weak association between Aβ deposition and disease severity - Invasiveness due to radiotracer injection - Variability in PiB uptake among healthy individuals - Limited understanding of Aβ accumulation in atypical AD presentations
Tau-PET Imaging	- Strong link between tau accumulation and cognitive decline - High affinity of radiotracers for PHF tau - Robust correlation between neurodegeneration and NFTs	- Invasiveness from radiotracer injection - Variability in tau morphology across AD subtypes - Ongoing evolution of this imaging modality in research and clinical practice

**Table 2 diagnostics-15-00612-t002:** Overview of the implemented XAI models using different imaging modalities with respective tasks and their accuracies.

XAI Methods	Imaging Modality	Tasks
LIME	MRI	Providing visual proof by highlighting image regions for prediction, utilizing images and gene expression data to classify AD and investigating specific brain regions responsible for classification, and investigating the apparent measures of diffusion MRI (dMRI) modality using reduced acquisitions for initial phases of AD
SHAP	MRI, PET	Investigating various modalities that influence the risk assessment of AD, selecting the most informative subjects, measuring the influence of each cognitive index on cognitive status, analyzing the most significant features for AD prognosis, predicting progression from MCI to AD, determining the order of informative predictors for diagnosing and prognosis, evaluating gender-related variations in forecasting the disease progression, investigating the dMRI measures for identifying the primordial phases of AD, exploring the multivariate connections between regional imaging metrics and cognitive function, distinguishing different dementia causes including AD, Lewy body-induced dementia, frontotemporal dementia, and healthy controls, and classifying AD severity
LRP	MRI, PET	Identifying critical regions of the cerebrum, visualizing relevant regions for positive AD classification, identifying the stages of AD, and finding neuropathological aspects of the AD
GradCAM	MRI	AD diagnosis, identifying potential irregularities in brain imaging and investigating the specific cerebral regions of interest (ROIs), classifying between healthy aging controls and MCI patients, classifying dementia into various categories, measuring the effectiveness of models against established AD biomarkers, and predicting the MCI-to-AD transition
SMs	MRI	Revealing the relevance of white matter hyperintensity lesions in AD-affected patients compared to healthy individuals, visualizing heatmaps that distinguish between healthy controls and AD, and early AD detection
GNNE	-	AD prognostic prediction based on longitudinal neuroimaging data from previously established datasets

**Table 3 diagnostics-15-00612-t003:** Overview of pros and cons of various XAI algorithms in AD neuroimaging.

Algorithm	Advantages	Disadvantages
LIME	- Model-agnostic - Provides local explanations - Relatively simple and intuitive method - Can be applied in image, text, and tabular data	- Local approximation may not fully capture model behavior - Explanations can be inconsistent across different runs - Does not provide global model understanding - Inability to capture nonlinear relationships - Does not consider collinearity
SHAP	- Provides more consistent and locally accurate explanations than LIME - Explains global model behavior and individual predictions - Can be applied in image, text and tabular data	- Computationally expensive - Explanations can be noisy for certain models - Does not consider collinearity in the original method
LRP	- Explains CNN models - Computationally efficient - Trustworthy and robust explanation method - Can be applied in both image and text data	- Specific to neural networks in the original method, not model-agnostic - Explanations can be noisy and lack intuition - Hyperparameters can significantly impact explanations
Grad-CAM	- Highlights important regions in images for predictions - Simple to implement for CNNs - Provides visual explanations that are easy to interpret	- Limited to CNNs and image data - Does not provide feature importance scores - Coarse explanations at image level, not pixel level
SMs	- Simple gradient-based technique for CNNs - Computationally efficient - Can be applied to both image and text data	- Lacks theoretical grounding compared to other methods - Sensitive to minor input perturbations - Focuses on pixel-wise importance (often noisy) - Very fragile and unreliable
GNNE	- Can model relational structure in data like molecules - Leverage graph convolutions and message passing - Achieve state-of-the-art performance on many graph learning tasks	- Complex architectures harder to train and interpret - Limited ability to transfer across graph domains - Lack of robust benchmarks and evaluation metrics

## Data Availability

We have included all relevant information in this article; if further clarification is required, please get in touch with the corresponding author.

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
