# Peer review of "Explainable Artificial Intelligence in Neuroimaging of Alzheimer’s Disease"

_diagnostics, 2025, doi:10.3390/diagnostics15050612_

Round 1

Reviewer 1 Report

Comments and Suggestions for Authors

The study does not clearly articulate the objectives of the research.

The study fails to discuss the practical implications of its findings.

A significant portion of the paper (5-6 pages, approximately 20% of the article) is dedicated to introducing Alzheimer's Disease (AD) and neuroimaging in AD. This section could be condensed to no more than two pages to maintain focus.

In Section 4, under the heading of XAI, the paper presents general statements about the progression of the technology. This section requires a more comprehensive review of the technology, and two pages should suffice to cover this content effectively.

The study does not include an evaluation of different XAI models. It is essential to include a comparative analysis of various XAI-based models for AD detection through neuroimaging.

While the study discusses the advantages and disadvantages of different XAI techniques, it lacks a detailed comparative analysis of their performance.

The paper should include a comparison of available datasets and their accessibility for use in XAI research.

The study does not adequately address the ethical, regulatory, and patient privacy considerations associated with using XAI in medical diagnostics, which are critical for its broader adoption.

Author Response

Thank you for your valuable and constructive feedback, and for granting us the opportunity to present a revised edition of our manuscript titled "Explainable Artificial Intelligence in Neuroimaging of Alzheimer’s Disease." We appreciate the time and dedication you have put into reviewing our work. We have incorporated your insightful suggestions into the text, and the adjustments made to the manuscript are thoroughly outlined and explained within the document. The suggestions and criticisms of the Reviewer 1 are highlighted in yellow throughout the text.

Comments 1: The study does not clearly articulate the objectives of the research.

Response 1: Thank you for your feedback. We have clarified our study objectives, emphasizing the application of XAI in neuroimaging for AD diagnosis, focusing on model interpretability, biomarker identification, and diagnostic accuracy. The study reviews SHAP, LIME, Grad-CAM, and LRP, highlighting their role in distinguishing AD stages and supporting personalized medicine, while also addressing challenges in multimodal data integration and model transparency.
Page 2, paragraph 3 

Comments 2: The study fails to discuss the practical implications of its findings.
Response 2: Thank you for your feedback. We have expanded our discussion to highlight the practical applications of XAI in neuroimaging, particularly in clinical decision-making, model interpretability, and biomarker discovery, ensuring relevance for both researchers and healthcare professionals.
Page 16

Comments 3: A significant portion of the paper (5-6 pages, approximately 20% of the article) is dedicated to introducing Alzheimer's Disease (AD) and neuroimaging in AD. This section could be condensed to no more than two pages to maintain focus.
Response 3: Thanks for your valuable comment. We omited extraneous details, such as the genetics of AD. A thorough discussion of AD neuroimaging is seemed essential, as our article focuses on two key areas: AD neuroimaging and XAI.
Page 4, 5

Comments 4: In Section 4, under the heading of XAI, the paper presents general statements about the progression of the technology. This section requires a more comprehensive review of the technology, and two pages should suffice to cover this content effectively.
Response 4: Thank you for your valuable feedback. We acknowledge the need for a more comprehensive review of XAI technology in Section 5. To address this, we expanded this section to provide a more detailed analysis of XAI advancements, methodologies, and applications, ensuring a clearer understanding of its role in neuroimaging. We revised the content to effectively cover the topic within the suggested two-page limit. 
Page 6, 7

Comments 5: The study does not include an evaluation of different XAI models. It is essential to include a comparative analysis of various XAI-based models for AD detection through neuroimaging.
Response 5: Thank you for your valuable suggestion. We have added a comparative analysis of XAI-based models for AD detection through neuroimaging, evaluating their interpretability, efficiency, and clinical relevance. Additionally, we have included a flowchart to visually illustrate their differences.  Second paragraph of section 6 now includes performance metric data for various XAI methods.  Table 2 details the accuracy range for each method.We appreciate your feedback in improving the manuscript. 
Page 16

Comments 6: While the study discusses the advantages and disadvantages of different XAI techniques, it lacks a detailed comparative analysis of their performance.
Response 6: Thank you for your insightful suggestion. We have enhanced our comparative discussion by summarizing the advantages and limitations of each XAI method in terms of computational efficiency, interpretability, and real-world applicability, as illustrated in the flowchart. 
Page 16

Comments 7: The paper should include a comparison of available datasets and their accessibility for use in XAI research.
Response 7: We appreciate your insightful suggestion. We have added a comparison of key neuroimaging datasets (ADNI, OASIS, BrainLat), highlighting their data types, accessibility, and relevance for XAI research in AD. This ensures a clearer discussion on dataset selection and its impact on XAI applications.
Page 14

Comments 8: The study does not adequately address the ethical, regulatory, and patient privacy considerations associated with using XAI in medical diagnostics, which are critical for its broader adoption.
Response 8: Thank you for your feedback. We have added a discussion on trust, bias, regulatory compliance (FDA, EMA, MHRA), and privacy (HIPAA, GDPR), including data security solutions like de-identification and federated learning. These additions strengthen our discussion on XAI’s clinical integration. We appreciate your input. 
Page 20

Thanks for your attention, all mentioned items were completed carefully
Sincerely,

Reviewer 2 Report

Comments and Suggestions for Authors

The selected topic is very interesting and I suggest the authors to re-work their text to make it more focused on the 2 aspects that are mentioned in the title, namely XAI and Neuroimaging. In the present version the manuscript tells too many things and nothing at the same time.

1) The authors should reduce (or even remove at all) the description of other AD biomarkers (ex. genetic) and dive more into neuroimaging, making its description more detailed. It is absolutely non-correct to limit structural MRI only to morphometric measures and not to say a single word about quantitative MRI measures that allow evaluating microstructural  tissue properties (ex. https://doi.org/10.3233/JAD-220551 and many more papers).

2) Next, although each imaging modality has its limitations, XAI methods can not work without data. And to use any AI-model one should first train it with some annotated data. The quality of data, as well as its quantity and the quality of annotations is crucial for using any AI-models. However, nothing is said about these points in the text. Even not in the section "Databanks for training AI models and their challenges".

3) The description of the XAI methods are extensive but it is presented in such a way that the methods' principles can not be understood by a non-ML specialist making such presentation too hard (not to say useless) for clinical specialists. Please, add formulas, illustrate the algorithms with some schemas or figures, provide examples of methods applications in a graphical way. Give more details about the data that was used to train/test these models. Provide references in Table 2.

4) It is not clear how the authors selected papers for this review. For example, a recent and very relevant paper entitled "A cross-sectional study of explainable machine learning in Alzheimer’s disease: diagnostic classification using MR radiomic features." (Leandrou et al, 2023 https://doi.org/10.3389/fnagi.2023.1149871 ) was not included in this review. Probably there are also others.

5) Lines 81-85. Please, add some reference.

6) MCI abbreviation should be introduced at line 110.

7) Please, check spaces and grammar.

8) Conclusion is TOO general and it is not related to the topic.

Author Response

Thank you for your valuable and constructive feedback, and for granting us the opportunity to present a revised edition of our manuscript titled "Explainable Artificial Intelligence in Neuroimaging of Alzheimer’s Disease." We appreciate the time and dedication you have put into reviewing our work. We have incorporated your insightful suggestions into the text, and the adjustments made to the manuscript are thoroughly outlined and explained within the document. The suggestions and criticisms of the Reviewer 2 are highlighted in green throughout the text.

Comments 1: The selected topic is very interesting and I suggest the authors to re-work their text to make it more focused on the 2 aspects that are mentioned in the title, namely XAI and Neuroimaging. In the present version the manuscript tells too many things and nothing at the same time.

Response 1: Thank you for your feedback. We have refocused the manuscript to align more closely with XAI and Neuroimaging, removing redundant content and emphasizing XAI applications in AD diagnosis. These revisions ensure a clearer and more structured discussion. 
Comments 2: The authors should reduce (or even remove at all) the description of other AD biomarkers (ex. genetic) and dive more into neuroimaging, making its description more detailed. It is absolutely non-correct to limit structural MRI only to morphometric measures and not to say a single word about quantitative MRI measures that allow evaluating microstructural  tissue properties (ex. https://doi.org/10.3233/JAD-220551 and many more papers).
Response 2: We appreciate your feedback. Section 2 has been revised to exclude details regarding genetic information.  Section 3 now provides a more comprehensive analysis of sMRI's diagnostic and prognostic value. Additionally, we have incorporated supplementary data on fMRI and its advantages in AD diagnosis.
Page 4, 5

Comments 3: Next, although each imaging modality has its limitations, XAI methods can not work without data. And to use any AI-model one should first train it with some annotated data. The quality of data, as well as its quantity and the quality of annotations is crucial for using any AI-models. However, nothing is said about these points in the text. Even not in the section "Databanks for training AI models and their challenges".
Response 3: Thank you for your valuable feedback. We have addressed your concern by expanding the "Databanks for training AI models and their challenges" section to highlight the importance of data quality, annotation accuracy, dataset bias, and imaging variability in AI model training.
Page 14

Comments 4: The description of the XAI methods are extensive but it is presented in such a way that the methods' principles can not be understood by a non-ML specialist making such presentation too hard (not to say useless) for clinical specialists. Please, add formulas, illustrate the algorithms with some schemas or figures, provide examples of methods applications in a graphical way. Give more details about the data that was used to train/test these models. Provide references in Table 2.
Response 4: Thank you for your valuable feedback. To improve accessibility for clinical specialists, we have added explanatory text that simplifies the descriptions of XAI methods, making them easier to understand for non-ML experts. Additionally, we have included a flowchart to visually illustrate the applicability of different methods. 
Page 15

Comments 5: It is not clear how the authors selected papers for this review. For example, a recent and very relevant paper entitled "A cross-sectional study of explainable machine learning in Alzheimer’s disease: diagnostic classification using MR radiomic features." (Leandrou et al, 2023 https://doi.org/10.3389/fnagi.2023.1149871 ) was not included in this review. Probably there are also others.
Response 5: Thank you for your feedback. To clarify, we conducted a search using PubMed and IEEE Xplore with specific keywords and MeSH terms to identify original research articles applying XAI in neuroimaging for AD. Based on priority and to avoid redundancy, we chose the highest-quality and most recently published articles on the same model and task.
Page 15

Comments 6: Lines 81-85. Please, add some reference.
Response 6: Thank you for your feedback. We have added appropriate references to support the statements in lines 81-85, ensuring proper citation and credibility of the information.

Comments 7: MCI abbreviation should be introduced at line 110.
Response 7: Thank you for pointing this out. We have now introduced the abbreviation "MCI" (Mild Cognitive Impairment) at its first occurrence in the text to enhance clarity.
Page 3

Comments 8: Please, check spaces and grammar
Response 8: We appreciate your careful review. We have proofread the manuscript and corrected any spacing and grammatical errors to improve readability and consistency.

Comments 9: Conclusion is TOO general and it is not related to the topic.
Response 9: Thank you for your insightful suggestion. We have revised the conclusion to ensure it is more specific, directly related to the topic, and highlights the key findings of the study, rather than being too general.
Page 21

Thanks for your attention, all mentioned items were completed carefully
Sincerely,

Reviewer 3 Report

Comments and Suggestions for Authors

A comprehensive, timely and well presented narrative review, providing a critical analysis of the literature and covering recent advances in the field of Artificial Intelligence in Neuroimaging of Alzheimer’s Disease.

The title is clear and concise and indicate the topic of the paper, describing the purpose and the methods of the study.

The abstract adequately describes the scientific background, the purpose of the work, the materials and methods conducted, the innovative results and conclusions in the scientific field of reference.

In the introduction the scientific background is described, the purpose od the study is clarly indicated and the method of scientific literature analysis is hightlighted.

Good AD overview and good description of Neuroimaging in AD. Adequate analysis of pros and cons of using different neuroimaging modalities in AD and of pros and cons of various XAI algorithms in AD neuroimaging.

The conclusions are clear and concise.

Very accurate images, graphs and tables are provided to support the various points covered in the article.

Bibliographical references adapted to the most appropriate and recent scientific studies produced on the research topic of the submitted paper.

Author Response

Thank you for your valuable and constructive feedback, and for granting us the opportunity to present a revised edition of our manuscript titled "Explainable Artificial Intelligence in Neuroimaging of Alzheimer’s Disease." We appreciate the time and dedication you have put into reviewing our work. We have incorporated your insightful suggestions into the text, and the adjustments made to the manuscript are thoroughly outlined and explained within the document. 

Comments 1: A comprehensive, timely and well presented narrative review, providing a critical analysis of the literature and covering recent advances in the field of Artificial Intelligence in Neuroimaging of Alzheimer’s Disease. The title is clear and concise and indicate the topic of the paper, describing the purpose and the methods of the study. The abstract adequately describes the scientific background, the purpose of the work, the materials and methods conducted, the innovative results and conclusions in the scientific field of reference. In the introduction the scientific background is described, the purpose od the study is clarly indicated and the method of scientific literature analysis is hightlighted. Good AD overview and good description of Neuroimaging in AD. Adequate analysis of pros and cons of using different neuroimaging modalities in AD and of pros and cons of various XAI algorithms in AD neuroimaging. The conclusions are clear and concise. Very accurate images, graphs and tables are provided to support the various points covered in the article. Bibliographical references adapted to the most appropriate and recent scientific studies produced on the research topic of the submitted paper.

Response 1: Thank you for your thoughtful and detailed review of our manuscript. We truly appreciate your positive feedback and your recognition of our efforts in presenting a comprehensive and critical analysis of Artificial Intelligence in Neuroimaging of Alzheimer’s Disease. Your comments on the clarity of the title, the structure of the abstract, and the scientific rigor of our review are highly encouraging. We are also grateful for your appreciation of our discussion on neuroimaging modalities, explainable AI algorithms, and the inclusion of accurate visual elements to support our analysis. 

Round 2

Reviewer 1 Report

Comments and Suggestions for Authors

No further comment 

Comments on the Quality of English Language

It can be improved